# Knowledge syntheses in medical education: Meta-research examining author gender, geographic location, and institutional affiliation

**Lauren A. Maggio**[1]*, **Anton Ninkov**[2], **Joseph A. Costello**[3], **Erik W. Driessen**[4], **Anthony R. Artino, Jr.**[5]

**1** Department of Medicine, Uniformed Services University of the Health Sciences in Bethesda, Bethesda, Maryland, United States of America, **2** School of Information Studies, University of Ottawa, in Ottawa, Ontario, Canada, **3** Henry Jackson Foundation, Uniformed Services University of the Health Sciences, Bethesda, Maryland, United States of America, **4** Department of Educational Development & Research, Maastricht University, Maastricht, The Netherlands, **5** Department of Health, Human Function and Rehabilitation Sciences, The George Washington University School of Medicine and Health Sciences, Washington, DC, United States of America

* lauren.maggio@usuhs.edu

**Data Availability Statement:** All relevant data are within the manuscript and its Supporting Information files.

## Abstract

### Introduction

Authors of knowledge syntheses make many subjective decisions during their review process. Those decisions, which are guided in part by author characteristics, can impact the conduct and conclusions of knowledge syntheses, which assimilate much of the evidence base in medical education. To better understand the evidence base, this study describes the characteristics of knowledge synthesis authors, focusing on gender, geography, and institution.

### Methods

In 2020, the authors conducted meta-research to examine authors of 963 knowledge syntheses published between 1999 and 2019 in 14 core medical education journals.

### Results

The authors identified 4,110 manuscript authors across all authorship positions. On average there were 4.3 authors per knowledge synthesis (SD = 2.51, Median = 4, Range = 1–22); 79 knowledge syntheses (8%) were single-author publications. Over time, the average number of authors per synthesis increased (M = 1.80 in 1999; M = 5.34 in 2019). Knowledge syntheses were authored by slightly more females (n = 2047; 50.5%) than males (n = 2005; 49.5%) across all author positions. Authors listed affiliations in 58 countries, and 58 knowledge syntheses (6%) included authors from low- or middle-income countries. Authors from the United States (n = 366; 38%), Canada (n = 233; 24%), and the United Kingdom (n = 180; 19%) published the most knowledge syntheses. Authors listed affiliation at 617 unique institutions, and

**Funding:** The authors received no specific funding for this work.

**Competing interests:** The authors have declared that no competing interests exist.

first authors represented 362 unique institutions with greatest representation from University of Toronto (n = 55, 6%). Across all authorship positions, the large majority of knowledge syntheses (n = 753; 78%) included authors from institutions ranked in the top 200 globally.

## Conclusion

Knowledge synthesis author teams have grown over the past 20 years, and while there is near gender parity across all author positions, authorship has been dominated by North American researchers located at highly ranked institutions. This suggests a potential over-representation of certain authors with particular characteristics, which may impact the conduct and conclusions of medical education knowledge syntheses.

## Introduction

Medical education is postsecondary education related to the practice of medicine; it typically includes: (a) initial training to become a physician (i.e., medical school and internship), (b) follow-on graduate medical education (i.e, residency and fellowship), and (c) continuing professional development. In medical education, researchers have been encouraged to publish knowledge syntheses and educators to act as evidence-informed practitioners in their application of these reviews toward the education of medical trainees and practicing physicians [1,2]. As a result, the recent proliferation of knowledge syntheses published in core medical education journals is unsurprising [3]. Knowledge syntheses, which often form the evidence base for implementing curricular innovations and determining how a field defines its key terminology, can have immense impact on a field's discourse and future directions [4,5].

The Canadian Institutes of Health Research (CIHR) defines knowledge syntheses (aka, reviews) as: "the contextualization and integration of research findings of individual research studies within the larger body of knowledge on the topic." [6] When conducting a knowledge synthesis, author teams are required to make multiple decisions, many of which can be subjective. For example, authors must decide which populations to include and which to exclude; which contexts matter and which do not; which factors are important to extract from the primary studies and which are not; and even which languages to review and which to exclude. Such decisions are shaped by author characteristics, backgrounds, and even the power structures and cultural norms in which the authors operate. Therefore, best practices in knowledge syntheses encourage scholars to assemble a diverse author team with representation from a variety of backgrounds and perspectives [7–9]. And these factors have the potential to impact the conduct of any given knowledge synthesis, as well as the conclusions drawn from the analysis [10]. In short, author characteristics have important implications for a field's evidence base.

Broadly speaking, researchers have raised concerns that author characteristics, such as gender [11–14], geographical location [15–17], and institutional affiliation [18,19], can bias publications, including knowledge syntheses, and inadvertently reinforce dominant power structures. To this point, the Cochrane Collaboration, a major supporter and publisher of systematic reviews, has flagged the lack of international representation and diversity in published reviews as a significant problem and reports that more diverse author teams generate more relevant reviews with less research waste and fewer errors [20]. In 2020, based in part on these findings, the Cochrane Collaboration advocated for wide participation from a variety of stakeholders in conducting reviews as one of its key strategic initiatives [21].

In medical education, we know little about the characteristics of the authors who write knowledge syntheses, and so we lack a clear understanding of which author voices dominate and which are absent from the evidence base created through these reviews. This gap in our understanding means we risk inadvertently prioritizing certain views while diminishing others, and potentially creating an evidence base that is irrelevant to some people in some contexts. For example, a review written by a United States author team on providing student feedback may resonate well with North American readers. However, due to a series of author decisions (e.g., inclusion criteria) and interpretations that likely vary based on cultural norms, such a review may be less useful to an audience outside of North America where cultural norms and educational systems differ.

Medical education researchers have just begun examining authorship characteristics. For example, a recent study explored author gender from articles published in four medical education journals [22], and another investigated authors' geographic location in papers indexed as medical education [23]. While both of these recent studies are valuable, they do not specifically examine knowledge syntheses, which have become a critical part of the evidence base in medical education.

The purpose of this study is to describe and examine the characteristics of the authors of knowledge syntheses with a focus on gender, geographical location, and institutional affiliation. In doing so, we hope to raise medical educators' awareness of author characteristics that may have bearing on the current state of the field's evidence base. We also hope to foster a healthy skepticism in the medical education evidence, which has been and continues to be synthesized through review articles.

## Materials and methods

We conducted meta-research to examine the authors of knowledge syntheses published in 14 core medical education journals between 1999–2019.

### Data collection

To undertake this meta-research, we utilized a publicly accessible data set that members of this author team created in 2020 and is licensed under a CC BY 4.0 license [24]. The data set includes citations and related PubMed and Web of Science (WoS) metadata for 963 knowledge syntheses published in 14 core medical education journals between 2009–2019 (See S1 Appendix for complete journal list and search strategies). The 14 journals were selected based on previous bibliometric studies that had identified these titles as "core" based on their presence in WoS and their perceived relevance to medical education by members of the field [25,26]. Additionally, all journals are indexed in PubMed/MEDLINE and have been publishing medical education research for over a decade. In the data set, we identified knowledge syntheses by screening the titles and abstracts of 2,210 articles published in these journals for articles that met the above CIHR definition of a knowledge synthesis. Full details on how we created the data set are published in Maggio et al., 2020 [3]. We utilized this existing data set because data reuse has been associated with reduced research waste, faster translation of research findings into practice, and enhanced reproducibility and transparency of science [27,28]. We chose this data set because it is the only existing, up-to-date data set of knowledge syntheses in medical education. All data were downloaded and managed in GoogleSheets [29].

To predict author gender, we extracted the first names of all authors from the data set. In cases where authors used initials only (e.g., D.A.D.C Jaarsma, aka Debbie Jaarsma), we conducted a web search to identify their first name. All first names were then submitted to the gender prediction tool Genderize.io [30]. Genderize.io predicts whether a name is male or female

based on a database of over 20,000 names and provides a probability that the name is either male or female. This tool has been used in multiple publications with similar aims to the current study (e.g., Hart and Perlis; [31] Bagga et al. [32]). We accepted the tool's designation for a name if the probability was over 70%. For each name that Genderize.io reported with <70% certainty (n = 151) and those 90 first names that the tool was unable to identify, we looked up the authors' online presence and cross referenced the authors' names with their publication and online profiles at their academic institutions and social media sites (e.g., LinkedIn, Academia. edu, ResearchGate) relying on author photos and the pronouns used in institutional bios. We recognize that our effort to predict gender is an oversimplification of a complicated social construct, especially because an individual's gender is best described by that individual. However, we believe this approach is a reasonable starting point to begin providing a sense of the field; it also follows the protocol of similar papers recently published on this topic [11,33].

### Geographical location

For each knowledge synthesis, we extracted from the WoS metadata the country of all author institutions. Due to the structure of the metadata, we were only able to accurately identify the location of the first author at the level of the individual author. Therefore, for non-first authors we report for each knowledge synthesis the countries represented in aggregate without regard to an individual's placement in author order. Thus, we can only make limited claims about a non-first author's geographical attribution. Countries were described using the World Bank's 2021 world region classification system [34], which includes four-levels of countries (low, lower middle, upper middle, and high income). Country levels are based on a country's gross national income [34].

### Institutions

For each knowledge synthesis, we identified the first author's institutional affiliation. Similar to geographical location, using WoS metadata we also identified for each knowledge synthesis a listing of all institutions that contributed to the study. To characterize institutional affiliation, we used the Times Higher Education (THE) World University Rankings for 2020 [35]. We selected this ranking due to its broad coverage of over 1,400 universities from 92 countries. THE ranking is based on 13 metrics (e.g., teaching, research, international outlook, industry outcomes). This ranking groups institutions starting with the top 200 institutions individually and then reports the remaining institutions in groups of 50, 100, or 200 depending on their position. In a minority of the cases examined in this study, authors provided multiple institutional affiliations (e.g., universities). In such cases, we included the first affiliation listed. Additionally, authors listed non-academic affiliations (e.g. professional associations, government entities, community hospitals), which we coded as organizations. Organizations are accounted for in our results, but unranked in relation to the THE rankings.

### Analysis

Descriptive statistics were calculated using GoogleSheets [29], and data were visualized using Tableau v.2020.04 [36].

### Results

We identified 4,110 authors listed across all authorship positions, and of those 3,200 were unique authors (See S2 Appendix for a listing of the most prolific authors). The number of authors per knowledge synthesis ranged from 1–22 with an average of 4.27 authors (SD = 2.51,

Median = 4). Seventy-nine knowledge syntheses (8.2%) were single-author publications. Over the 20-year time period examined, the average number of authors per knowledge synthesis increased (M = 1.80 in 1999; M = 5.34 in 2019; See Fig 1).

## Gender

We identified the gender for 4,052 authors. We were unable to make a confident prediction of gender identification for 59 authors (even after a web search), all of whom only appeared once in the dataset (See S3 Appendix for listing of author names and corresponding gender). Knowledge syntheses were authored by slightly more females (n = 2047; 50.5%) than males (n = 2005; 49.5%) across all author positions. In addition, more females were listed as first authors (n = 494; 51.9%) and second authors (n = 483; 55.4%). On the other hand, the last author position was held by more males (n = 404; 56.0%). See Fig 2 for author order by gender.

Most author teams were a combination of genders (n = 683 teams; 70.9%), but 280 teams included authors of a single gender (117 all female; 163 all male). For single authored papers, 52 were written by males and 26 by females. Over the time period examined, the ratio of female authors in all positions has increased (See Fig 3).

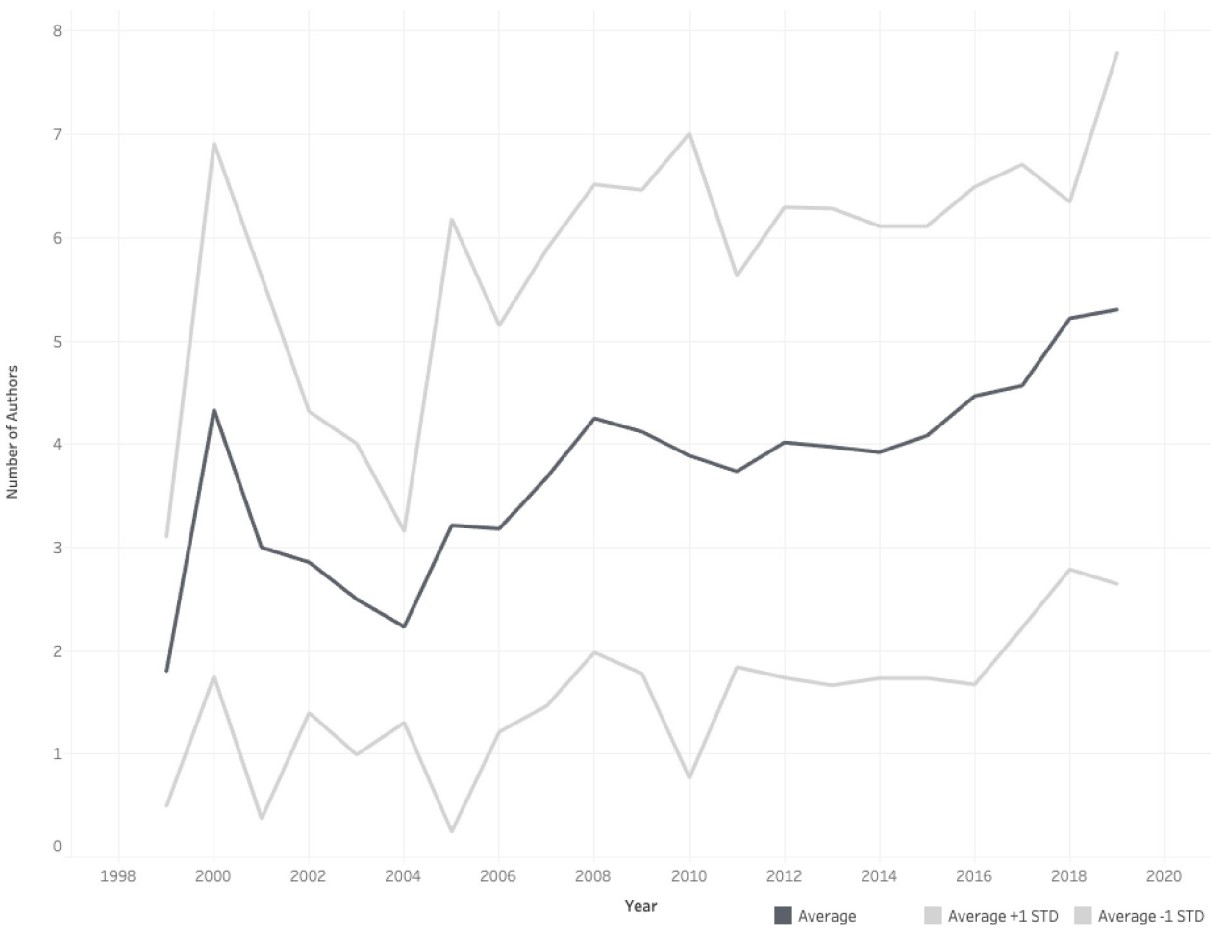

**Fig 1. The median number of authors per knowledge synthesis (10th and 90th percentiles also shown) published in 14 core medical education journals published between 1999–2019.**

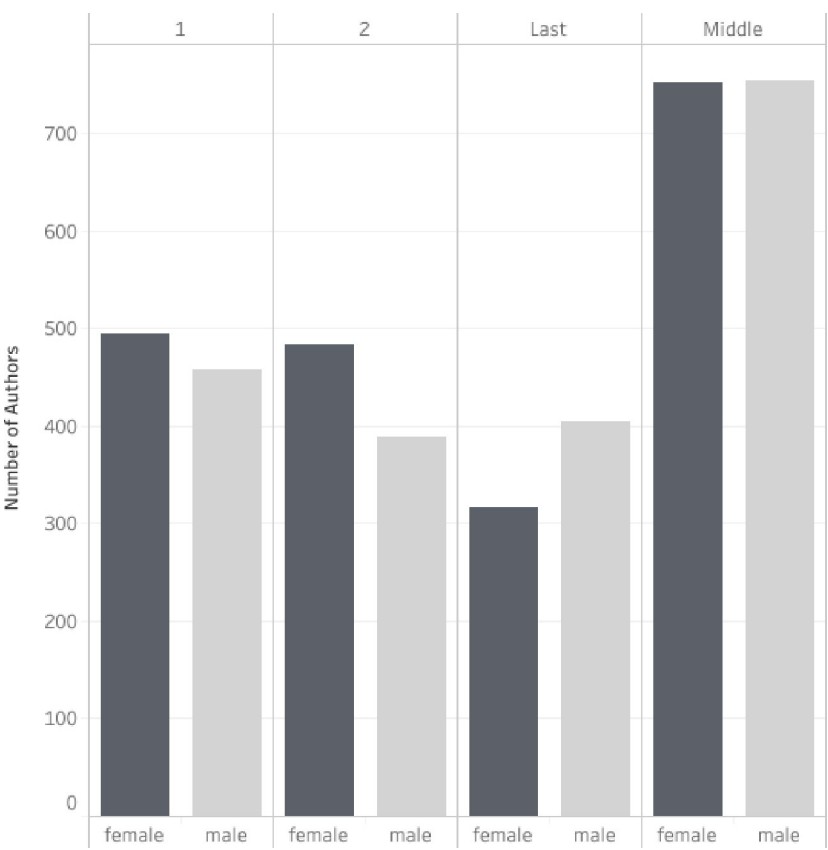

**Fig 2. Author order by gender for knowledge syntheses in 14 core medical education journals published between 1999–2019.** We were unable to determine the gender of 59 author names, which are excluded from this figure.

## Geography

Across all authorship positions, authors listed affiliations in 58 countries, including 22 LMIC (see Fig 4; See S4 Appendix for listing of all countries). By number of knowledge syntheses, the countries most represented were the United States (US) (n = 366; 38%), Canada (n = 233; 24%), and the United Kingdom (UK) (n = 180; 19%). Fifty-eight (6%) knowledge syntheses included at least one author listing an affiliation based in a LMIC; of these, 39 (4%) were first authored by an author with a LMIC affiliation. Of the 58 knowledge syntheses including LMIC authors, authors based in China were most prolific, publishing 22 knowledge syntheses of which 15 were written by authors who were all based in China. First authors represented 42 countries, including 13 LMIC. The most represented countries for first authors were the US (n = 312; 25%), Canada (n = 183; 15%), and the UK (n = 151; 12%). Twenty-seven (3%) knowledge syntheses were exclusively authored by individuals based in a LMIC. The most countries represented on a single team were seven, in a study featuring authors from France, Ireland, UK, Italy, Belgium, Croatia, and Germany [37]. Eighty percent (n = 767) of knowledge syntheses included authors from a single country only. Of those representing a single country, authors were predominantly located in the US (n = 271; 22%), Canada (n = 149; 12%), and the UK (n = 122; 10%).

## Institutions

Across all authorship positions, we identified 617 unique institutions (See S5 Appendix for complete list of institutions). Institutions most often represented were the University of

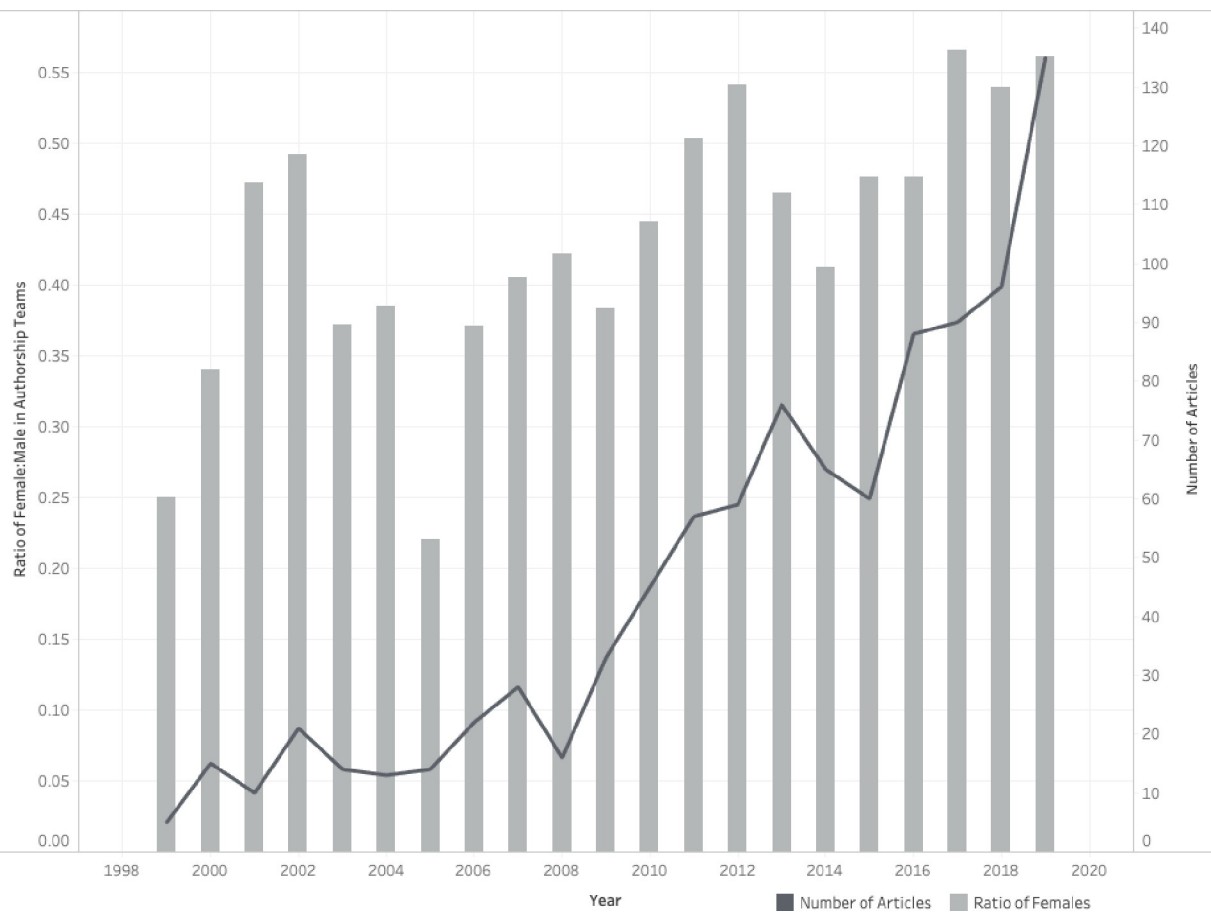

**Fig 3. The ratio of female authors in all authorship positions for knowledge syntheses published in 14 core medical education journals published between 1999–2019.**

Toronto (n = 212 authors), the Mayo Clinic (n = 110 authors), and Maastricht University (n = 89 authors). See Table 1 for the top 10 institutions by frequency. Nearly half (n = 451; 47%) of knowledge synthesis listed authors from a single institution. The most institutions represented on a knowledge synthesis was 14 [38].

For all authorship positions, 78% (n = 753) of knowledge syntheses included authors from institutions ranked in the THE top 200. The remaining knowledge syntheses included 368 with authors based at institutions ranked between 200–1000 and 217 at institutions ranked below 1000 or unranked. Of the 458 unique academic institutions represented, a total of 154 were unranked. Two hundred and twenty-nine authors (6%) represented non-university affiliated medical centers or hospitals and 229 (6%) listed affiliations at professional organizations.

First authors represented 362 unique institutions with the most first authors based at the University of Toronto (n = 56, 15%) and the Mayo Clinic (n = 31, 9%). The third most frequently represented institution was a tie between Monash University, McMaster University, University of Ottawa, and the National Health Service in the United Kingdom (n = 18; 5%). Of the top 200 institutions, only 110 (30%) institutions were represented in the first author position, yet the top 200 institutions accounted for 486 (50%) publications in our sample. Beyond the top 200, first authors represented 101 (28%) institutions ranked between 200–1000 and another 89 (25%)

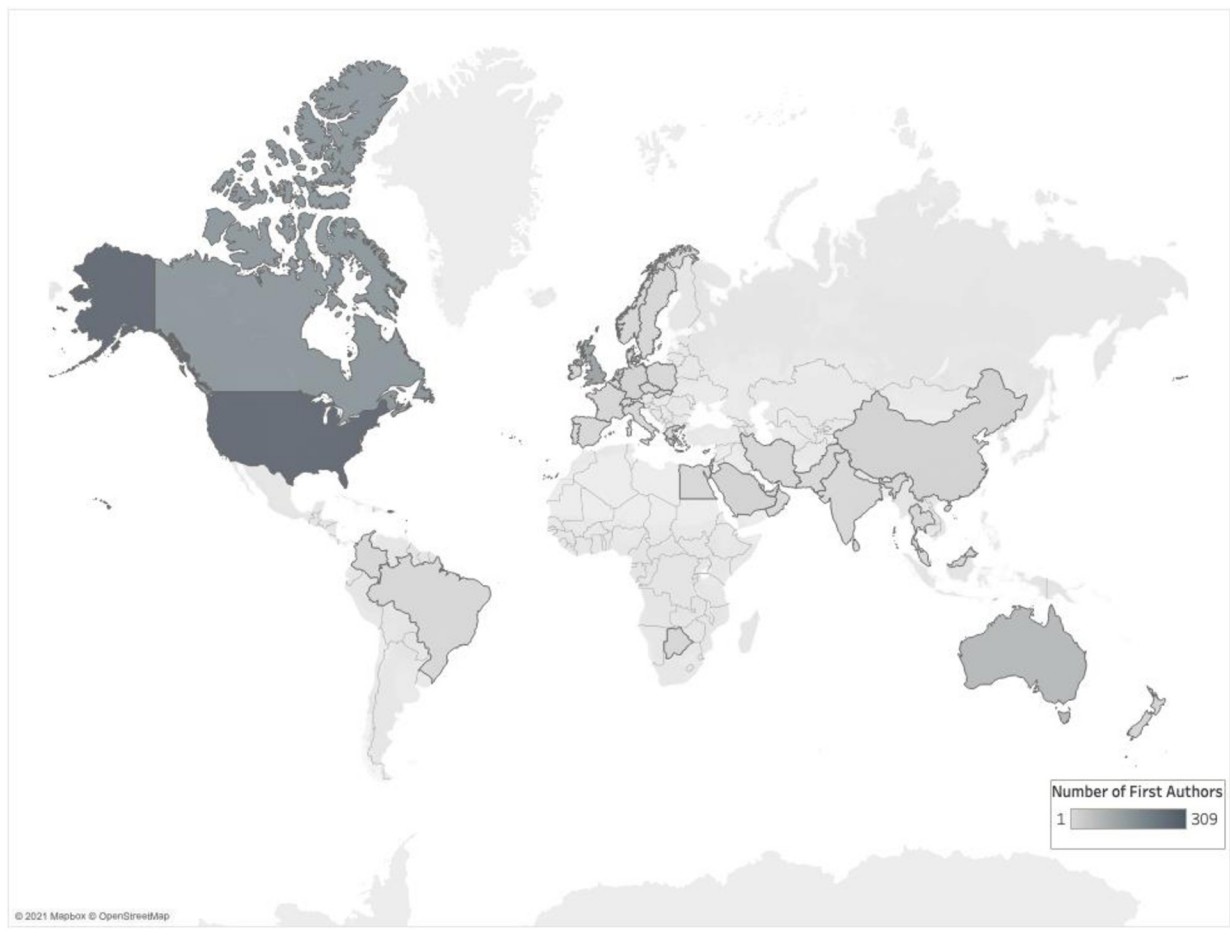

**Fig 4. Map highlighting the 42 countries listed as affiliations of first authors of knowledge syntheses published in 14 core medical education journals published between 1999–2019.** Contains information from OpenStreetMap and OpenStreetMap Foundation, which is made available under the Open Database License.

**Table 1. Top 10 institutional affiliations by count of first authors of knowledge syntheses published in a core set of medical education journals 1999–2019.**

| Institution (country) | Count of first authors (%) | Times Higher Education Ranking | Count of authors across all authorship positions (%) |
|---|---|---|---|
| University of Toronto (Canada) | 56 (5.7) | 18 | 212 (5.2) |
| Mayo Clinic (United States) | 32 (3.3) | ORG | 110 (2.7) |
| McMaster University (Canada) | 18 (1.9) | 72 | 64 (1.6) |
| Monash University (Australia) | 18 (1.9) | 75* | 72 (1.8) |
| University of Ottawa (Canada) | 18 (1.9) | 141* | 72 (1.8) |
| National Health Services (United Kingdom) | 17 (1.8) | ORG | 70 (1.7) |
| University of British Columbia (Canada) | 15 (1.6) | 34 | 52 (1.3) |
| University of Utrecht (Netherlands) | 14 (1.5) | 75* | 76 (1.8) |
| McGill University (Canada) | 13 (1.3) | 42 | 52 (1.3) |
| University of Alberta (Canada) | 13 (1.3) | 136* | 54 (1.3) |
| University of Calgary (Canada) | 13 (1.3) | 201–250 | 47 (1.1) |
| Maastricht University (Netherlands) | 13 (1.3) | 127 | 89 (2.2) |

Rankings retrieved from *Times Higher Education* (THE) World University Rankings 2020.

* indicates a tie in THE rankings.

ORG = organizations (e.g., health system).

institutions were beyond 1000 or unranked. Thirty-four (9%) first authors represented professional organizations and 25 (7%) non-university affiliated medical centers or hospitals.

## Discussion

In this meta-research, we described the author characteristics of 963 knowledge syntheses in medical education published between 1999–2019. We observed that the size of author teams has grown over the past 20 years, and while there is near gender parity across all author positions, authorship has been dominated by North American researchers located at highly ranked institutions.

The majority of knowledge syntheses examined here were multi-authored manuscripts, which aligns with the description of medical education researchers as inherently collaborative [39] and may reflect the significant time and effort required to publish a rigorous knowledge synthesis [40]. Notably, over the 20 years examined here, the average size of the author team increased. This growth mirrors a similar trend in science more broadly [41] and in some medical specialties more specifically [42,43]. Researchers have attributed this growth to multiple factors, including increased ease of collaboration between scholars as a result of computer and internet technology; utilization of research methodologies that require a variety of expertise and skills; the growing complexity of topics/research questions addressed; the overall increase of systematic reviews and meta-analyses; and the availability of guidelines for production of systematic reviews that require the inclusion of multiple authors [3,44,45]. In addition, findings from a study of research practices in medical education suggest that some of this growth in author team size may be related to questionable research practices like honorary authorship [46,47]. As the field of medical education continues to mature and its literature base grows, strategic selection of team members, including the rightsizing of the knowledge synthesis team, will become increasingly important. Thus, future research may be needed to further examine the size and composition of knowledge synthesis teams in order to provide evidence-based practical guidance on team construction.

The majority of author teams included both genders, suggesting that both male and female voices are present across the evidence being synthesized by these reviews. Over the study period, the ratio of females to males has increased. Researchers have attributed this growth to the increasing number of females entering medical school [48]. Additionally, gender parity among authorship teams has been achieved in recent years, which aligns with Madden's recent analysis of four medical education journals across a variety of publication types [22]. Furthermore, similar to the findings from Madden et al examining other medical specialities [49–51], we identified significantly more males in the last author position. In biomedicine, the last position is traditionally occupied by the "senior author" who takes on a leadership role in the study [52] or is often the principal investigator of the research laboratory conducting the work. Similarly, in medical education, the last author position is often held by the senior author. While further investigations are necessary, we speculate that this finding may be related to the under-representation of women in leadership positions in academic medicine [53]. Overall, however, the gender results observed in this study are encouraging; nonetheless, future work should continue to track author gender in medical education to monitor for additional changes that might occur over time. For example, recent research related to the impact of the COVID-19 pandemic on science raises concern that female investigators, especially those with younger children, have had less time for research [54] and writing and, as a result, may be publishing fewer papers during the pandemic than their male counterparts [55].

Although we identified author representation from 58 countries, 80% of author teams were based in a single country. Additionally, authorship was dominated by individuals based in the

US, Canada, and the UK, suggesting a heavy influence from English-speaking countries. This may have implications for the inclusion or potential exclusion of non-English language articles from reviews, as multinational teams are more likely to include non-English studies in their knowledge syntheses [56]. The exclusion of non-English articles is a known issue in the conduct of knowledge syntheses and has been labeled the "Tower of Babel Bias"; [57] this bias has implications for the accuracy and generalizability of research findings [58]. While examining the language inclusion criteria of each article is beyond the scope of this study, our findings suggest further investigation is warranted to better understand if the Tower of Babel Bias is an important issue in medical education knowledge syntheses.

Few authors listed affiliations in LMIC, and there was even less representation from LMIC in the first authorship position. This finding indicates geographical diversity is lacking in medical education knowledge syntheses, which has implications for the relevance of these reviews. In a 2019 study with similar findings to the present investigation, Thomas concluded that medical education research, more than any other field, is conducted by authors in the English-speaking Western countries, which he referred to as the "realm of the rich" [59]. This dominance of authors based in Western countries may limit the utility of these findings for non-Western health and education systems. To address some of this imbalance, the Cochrane Collaboration suggests that knowledge syntheses authors "take account of the needs of resource-poor countries and regions in the review process and invite appropriate input on the scope of the review and the questions it will address." [60]

Researchers have identified that knowledge syntheses conducted across multiple institutions can improve the quality and visibility of a publication, as well as help to avoid a "silo effect" [61,62]. Our findings demonstrate that just under half of the knowledge syntheses examined were multi-institutional investigations. Moreover, across all authorship positions, 78% of knowledge syntheses included authors affiliated with institutions ranked in the THE Top 200. As other medical education researchers have noted "The big players moreover are in a position to influence the global discourse more than others" [63]. As such, the field would do well to consider growing the number of multi-institutional collaborations that not only perform original research, but also collaborate to conduct knowledge syntheses.

## Limitations and future directions

There are a number of important limitations in the present work that suggest some fruitful areas of future research. First, our data set is composed of 14 core journals, which did not include journals from specific world regions, such as the *African Journal of Health Professions Education*. This limitation is particularly important, especially because we wanted to understand who does and does not have a voice in the development of knowledge syntheses. Had the data set examined other journals, we would likely have attained different results. That said, these 14 journals have been defined earlier as core medical education publications [25,26].

Second, we used a gender prediction tool to determine whether a first name was characterized as male or female. We recognize that this binary approach is an important limitation of our study and that, as noted above, an individual's gender is best described by that individual. Based on a survey of the literature for similar studies [11,33], we believe a critical need for future research is work that aims to more accurately ascertain investigator gender.

Third, we did not review the full texts of the knowledge syntheses we analyzed. Therefore, we are unable to make any claims about how author characteristics may have impacted the formulation of their research question, conduct of the knowledge synthesis, or their conclusions. Future work should consider a more in-depth examination of the full text to examine components like inclusion and exclusion criteria and the stated rationale behind the authors'

decisions. Additionally, researchers might consider investigating the full text of reviews to determine whether there is a difference between knowledge syntheses written, for example, by a global team of researchers in comparison to reviews conducted at a single institution. This follow-on work might also include qualitative inquiry to better understand how authors approached their review, including reflections on how their backgrounds may have impacted the conduct of the review. In relation to THE ranking, not all universities submit data for ranking and thus institutions may have been missed. Additionally, we identified authors from associations and organizations, which would not have been ranked, but that may have influence (e.g., the Association of American Medical Colleges).

## Conclusion

The production of knowledge syntheses, like all knowledge production, can be influenced by the authors' characteristics, backgrounds, and the power structures and cultural norms from which they operate [10,20]. In this study, we identified and critically examined the characteristics of the authors of knowledge syntheses to better understand the voices present–and those that may be missing–in the medical education evidence base. While gender parity has improved in recent years, knowledge synthesis authors predominantly work in elite institutions from high-income countries. Although more research is needed to truly understand the impact of these and other author characteristics, we suspect that some of the imbalances observed herein may have negative implications for medical education's evidence base and its global relevance.

## Supporting information

**S1 Appendix. List of core journals and search strategy.**
(DOCX)

**S2 Appendix. Most prolific authors.**
(DOCX)

**S3 Appendix. Author gender.**
(XLSX)

**S4 Appendix. All countries.**
(DOCX)

**S5 Appendix. Institutions.**
(DOCX)

## Author Contributions

**Conceptualization:** Lauren A. Maggio, Joseph A. Costello, Erik W. Driessen, Anthony R. Artino, Jr.

**Data curation:** Lauren A. Maggio, Anton Ninkov, Joseph A. Costello.

**Formal analysis:** Lauren A. Maggio, Anton Ninkov, Joseph A. Costello, Anthony R. Artino, Jr.

**Investigation:** Lauren A. Maggio, Anton Ninkov, Joseph A. Costello, Anthony R. Artino, Jr.

**Methodology:** Lauren A. Maggio, Anton Ninkov, Joseph A. Costello, Erik W. Driessen, Anthony R. Artino, Jr.

**Project administration:** Lauren A. Maggio.

**Resources:** Joseph A. Costello.

**Visualization:** Anton Ninkov, Joseph A. Costello.

**Writing – original draft:** Lauren A. Maggio, Anton Ninkov, Joseph A. Costello, Erik W. Driessen, Anthony R. Artino, Jr.

**Writing – review & editing:** Lauren A. Maggio, Anton Ninkov, Joseph A. Costello, Erik W. Driessen, Anthony R. Artino, Jr.

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
