## [Decision Letter · Decision Letter 0]

6 Jul 2021

PONE-D-21-18264

Knowledge syntheses in medical education: Examining author gender, geographic location, and institutional affiliation

PLOS ONE

Dear Dr. Maggio,

Thank you for submitting your manuscript to PLOS ONE. After careful consideration, we feel that it has merit but does not fully meet PLOS ONE’s publication criteria as it currently stands. Therefore, we invite you to submit a revised version of the manuscript that addresses the points raised during the review process.

The manuscript presents interesting data and is of potential interest. However, there are several important limitations that preclude to accept the manuscript in its current form. We invite you to submit a revised version of the manuscript that addresses the items indicated by the reviewers, with a point-by-point responses to their comments. In addition, I also have several suggestions as the Academic Editor:

1. Please provide more details about the manual classification names that Genderize.io reported with <70% certainty (n=151) and failed to identify (n=90) - which gender suggested Genderize (based on the highest probability) and which you identified.

2. Please extend the description of Institutions identification in the "Methods". Particularly, include the information on how many Institutions we not in the Times Higher Education World University Rankings. One of the reviewers pointed out a very important considerations about management of Institutions information by PubMed before and after 2013, this certainly should be described in the "Methods" and considered in the manuscript.

3. Please report the exact PubMed search strategy (in Appendix) by which the dataset has been obtained.

4. Please clarify the phrase "Therefore, for authors other than the first author, we report for each knowledge 172 synthesis all countries represented without regard to an individual’s placement in author 173 order." and explain how this influenced the geographic attribution for the non-first authors.

5. The statistically significant difference you reported in the phrase "Knowledge syntheses were authored by slightly more females (n=2047; 50.5%) than males (n=2005; 49.5%) across all author positions (Pearson Χ2=22.02, p<.001)." seems rather surprising taking into account the small absolute differences. In the "Methods" you only reported the tools used, but have not mentioned the statistical test applied. Please report statistical tests in details. Considering the use of p-value, it is allowed in the PLoS, but please report also the measurement of uncertainty where needed.

6. Please explain more in details which limitations could be related to the fact that your "research team is made up of individuals from the US, Canada, and the  Netherlands", why it is ironic and privileged, and which "various world perspectives that may be relevant in understanding important author characteristics" are reported in the world literature from other regions.

7. At the Figure 1 you reported "Average" number of authors, indicating the mean. Apart of the terminology, the parametric description does not fit well with the highly skewed distribution of number of authors. Please use instead the median and percentiles (the exact percentile is upon your choice) to present these findings.

8. Please provide more information about the Core medical education journals listed in the Supplemental Appendix A, with the major aim to justify their selection for the analysis.

9. Considering that the gender is one of the major topics of your manuscript, please present full first names of the authors included in the analysis and listed in the Suppl B.

10. You have performed an excellent and very elegant work. The vast majority of classifications have no doubts. However, there are several cases that looks to be classified in a different way. In the Appendix C you have provided full list of names and classifications (and I congratulate you with the adherence to the open research and full reporting!), and there a person having the name "Ashley" is classified as "No data on gender". There are oriental names that you classified as one-gender, while they used for both male and female name - for example Xiao (google the "Xiao male or female name") or Su (google the same or see https://genderchecker.com/pages/search-engine). I understand that identification of the gender could be made with knowing additional details you explained in the "Methods" (Linkedin search, etc) and that you can identify the exact gender for a given author with unisex name. Considering this, could you please introduce another column and indicate in the Appendix C for each author whether the gender was identified by Genderize.io or by manual search.

11. Please provide some explanations in the tables' footers to make the tables easier to interpret, for example the table in the Appendix E has a column "Count of KS across all author positions" that requires some brief explanations in the appendix (without a need to look at the main manuscript).

12. The figure 3 is very nice, and represents interesting findings. It would be worth to consider in the "Discussion" the possible reasons for the steep increase in female:male ratio during the short period from 2009 to 2012, and indicate in the "Results" whether these increase has been detected in all groups of countries, or limited only to high-income countries.

We look forward to receiving your revised manuscript.

Kind regards,

Boris Bikbov

Academic Editor

PLOS ONE

Journal Requirements:

2. We noted in your submission details that a portion of your manuscript may have been presented or published elsewhere.

[To undertake this study, we utilized a publicly accessible data set (http://doi.org/10.5281/zenodo.3990481) The data set includes citations and related PubMed metadata for 963 knowledge syntheses published in 14 core medical education journals between 2009-2019. We utilized this existing data set because data reuse has been associated with reduced research waste, faster translation of research findings into practice, and enhanced reproducibility and transparency of science.

In this present study, we used this publicly available data set in an earlier published study, which broadly described the included knowledge syntheses, but did not delve into authorship. We have attached this earlier manuscript to our submission. In this present study, we further enriched the existing data set by linking it with other resources, including genderize.io, the Times Higher Education Rankings, and the World Bank's World Region classification.

We do not believe this present submission constitutes dual publication because we designed this study with different research aims and enriched the existing data set to meet those aims. ]

Please clarify whether this publication was peer-reviewed and formally published. If this work was previously peer-reviewed and published, in the cover letter please provide the reason that this work does not constitute dual publication and should be included in the current manuscript.

3. We note that Figure 4 in your submission contain map images which may be copyrighted. All PLOS content is published under the Creative Commons Attribution License (CC BY 4.0), which means that the manuscript, images, and Supporting Information files will be freely available online, and any third party is permitted to access, download, copy, distribute, and use these materials in any way, even commercially, with proper attribution. For these reasons, we cannot publish previously copyrighted maps or satellite images created using proprietary data, such as Google software (Google Maps, Street View, and Earth). For more information, see our copyright guidelines: http://journals.plos.org/plosone/s/licenses-and-copyright.

1.    You may seek permission from the original copyright holder of Figure 4 to publish the content specifically under the CC BY 4.0 license. 

Reviewers' comments:

Reviewer's Responses to Questions

**Comments to the Author**

1. Is the manuscript technically sound, and do the data support the conclusions?

Reviewer #1: Yes

Reviewer #2: No

2. Has the statistical analysis been performed appropriately and rigorously? 

Reviewer #1: No

Reviewer #2: I Don't Know

3. Have the authors made all data underlying the findings in their manuscript fully available?

Reviewer #1: Yes

Reviewer #2: Yes

4. Is the manuscript presented in an intelligible fashion and written in standard English?

Reviewer #1: Yes

Reviewer #2: Yes

5. Review Comments to the Author

Reviewer #1: I have been invited to review the manuscript entitled “Knowledge syntheses in medical education: Examining author gender, geographic location, and institutional affiliation”. I appreciate the opportunity and thank the editors for considering me.

Overall, I found the article interesting, well-written, and well-informed. Therefore, to start with, I do recommend its publication. Please consider my following comments as an opportunity to improve the manuscript and introduce more clarity where I believe it needs it.

TITLE

I find the study design lacking in the title, which had me second-guessing for quite a while what was done. To avoid this, I suggest including the term “meta-research”. My suggestion, then, would be as follows:

“Knowledge syntheses in medical education: A meta-research examining author gender, geographic location, and institutional affiliation”

I don’t think it necessary to reference the prolific output by John Ioannidis on the topic of meta-research, but if the authors wish to follow up on this, there are several good articles published in Plos One by him.

METHODS

The authors say they have done a case study. While this is not incorrect, I think a better description of the study design is meta-research or research on research.

Lines 131-132

I think you should state clearly from the start that the 14 core journals you used are all indexed in PubMed/MEDLINE, making the “PubMed metadata” phrase a bit less cryptic for readers less versed in bibliometrics.

Lines 143-145

The sentence “Specifically,…” is not supported. Please delete. Additionally, it is repetitive.

Line 147

“Data enrichment” sounds strange to my ears. Why not “variables”?

Lines 158-161

Please consolidate the information provided for the 151 uncertain genders with the 90 not identified, as the same approach was used.

Lines 171-173

I did not understand this sentence.

Lines 186-187

When you use the expression “multiple affiliations,” please specify that you refer to multiple institutional affiliations (as in universities).

Lines 193

An analysis plan is lacking. In my view, this is a strictly descriptive study. You did not do a pre hoc sample size calculation. You used the whole population of knowledge synthesis articles for a set of journals, which is fine. However, you cannot make any generalizations from your data. You should only use descriptive statistics. You have no hypothesis to test.

RESULTS

Please remove any p-values from your results.

A comment on Geography. You say in methods that you will use the World Bank world region classification system based on the country’s gross national income. I think this indicator is profoundly misleading. Lacking a better one to offer, I understand that you would use it. But maybe consider including in the discussion that many HICs are profoundly different from the countries most represented in your findings (not unsurprisingly, US, Canada, and the UK). Many HICs are more like your LMIC. I consider Global South and Global North a more meaningful description of these differences; although inaccurate, it does place a key emphasis on power relations and asymmetries.

Line 284

Please specify the country of the National Health Service mentioned.

DISCUSSION

Please include a paragraph summing up the main findings before beginning the discussion.

Line 339

Should say “author” not “authors”.

Line 356

Reference [20] is to the Cochrane handbook and seems out of place here. I would suggest deleting the whole sentence on waste and bias as it distracts readers from the main thrust of the discussion: the concentration of knowledge synthesis in medical education in some (few) countries (that have in common not only English but also wealth).

Lines 394-396

Author characteristics not only impact conduct and conclusions but, first, and most importantly, the research questions.

Reviewer #2: Your article is fluent and well-organized. Your figures are clear and helpful. However, in their current form, your manuscript and the supporting appendices do not supply adequate information about your dataset. I am hoping that you checked all the affiliation data with the article webpages and just failed to make this clear in the manuscript. Based solely on the information you have supplied, I would believe that the author affiliation information in your dataset, for most of the journals studied, was drawn entirely from PubMed. You do not comment on the fact that MEDLINE indexing of authors' affiliations changed in 2013 or describe how you dealt with this change (see "Author, Corporate Author, and Collaborator Affiliation Display Changes." NLM Tech Bull. 2013 Nov-Dec;(395):e9.) . As you probably know, records indexed prior to 2013 were edited by the indexer to include the first author's affiliation only. For example, -- Cook DA, Bordage G, Schmidt HG. "Description, justification and clarification: a framework for classifying the purposes of research in medical education." Med Educ. 2008 Feb;42(2):128-33. doi: 10.1111/j.1365-2923.2007.02974.x. Epub 2008 Jan 8. PMID: 18194162. -- is indexed with Cook's Mayo Clinic affiliation only. Bordage's University of Illinois at Chicago affiliation and Schmidt's Erasmus University affiliation are not reflected in the PubMed record. If your affiliation data for most journals studied were derived solely from PubMed records, your data collection for 1999-2012 articles is seriously flawed. Even PubMed affiliation data for late 2013 - 2019 publications are not entirely trustworthy. The affiliation data are entered by the publisher and are not edited by PubMed's indexers. If the publisher enters the first author's affiliation only, that is the only affiliation reflected in PubMed. For your study, use of records from a literature database that focuses on affiliation indexing (like Scopus) would have made more sense. For an article about the effect of the indexing change on affiliation studies, see Ibarra, M.E., Ferreira, J.P., Torrents, M. et al. "Changes in PubMed affiliation indexing improved publication identification by country." Scientometrics 115, 1365–1370 (2018). https://doi.org/10.1007/s11192-018-2714-x

My other concerns are minor and/or philosophical and would not be sufficient to prevent publication. You frequently refer to the Cochrane Handbook. I feel that the Cochrane Handbook would be a more appropriate source of guidance if you were discussing reviews of therapeutic interventions and diagnostic techniques, the types of reviews published in the Cochrane Database of Systematic Reviews. Representation of research from around the world is relevant to reviews of therapeutic interventions and diagnostic techniques. As you note, educational techniques may be more culture-specific. You mentioned the fact that appropriate feedback methods vary depending on the culture. Is this, perhaps, a point in favor of production of culture-specific syntheses rather than a point in favor of including all voices in a given article? Authors and readers want to focus on information that is relevant in their setting. If author/reader time and article length were unlimited, many educators would want to know everything about all cultures and educational systems, but sadly both time and article length are typically limited. This brings me to my next point. The Cochrane Handbook would be a more appropriate guide if you were discussing articles in journals with essentially no article length limits. As articles in the Cochrane Database of Systematic Reviews are often very lengthy, there is room in these reviews to explore the delivery of care in nations with different economic resource levels. I do agree that information and voices from around the world are relevant to many educational research topics. I also think that many reviews must be focused on educational systems and/or cultures similar to the authors' to accomplish the authors' objectives within many journals' word-number limits ( even when awareness of other cultures and systems would be helpful to educators who deal with students who have come from other cultures).

You mention the growth in number of authors per article and provide some possible reasons for this growth. You haven't mentioned the rapid growth in the publication of systematic reviews and meta-analyses as one reason for this change. Guidelines for production of systematic reviews/meta-analyses require the participation of a multi-author team.

Your citation 20 on line 97 refers to the entire Cochrane Handbook. It would be better to cite specific chapters, as you have in other in-text citations. Your reader should not have to comb the entire Handbook to find the source of your statement.

6. PLOS authors have the option to publish the peer review history of their article (what does this mean?). If published, this will include your full peer review and any attached files.

Reviewer #1: **Yes: **Vivienne C. Bachelet

Reviewer #2: No

---

## [Author Response · Author response to Decision Letter 0]

14 Sep 2021

Please see the attached revision table.

---

## [Decision Letter · Decision Letter 1]

11 Oct 2021

Knowledge syntheses in medical education: Examining author gender, geographic location, and institutional affiliation

PONE-D-21-18264R1

Dear Dr. Maggio,

We’re pleased to inform you that your manuscript has been judged scientifically suitable for publication and will be formally accepted for publication once it meets all outstanding technical requirements.

Kind regards,

Boris Bikbov

Academic Editor

PLOS ONE

Additional Editor Comments (optional):

Reviewers' comments:

Reviewer's Responses to Questions

**Comments to the Author**

1. If the authors have adequately addressed your comments raised in a previous round of review and you feel that this manuscript is now acceptable for publication, you may indicate that here to bypass the “Comments to the Author” section, enter your conflict of interest statement in the “Confidential to Editor” section, and submit your "Accept" recommendation.

Reviewer #1: All comments have been addressed

Reviewer #2: All comments have been addressed

2. Is the manuscript technically sound, and do the data support the conclusions?

Reviewer #1: Yes

Reviewer #2: Yes

3. Has the statistical analysis been performed appropriately and rigorously? 

Reviewer #1: Yes

Reviewer #2: Yes

4. Have the authors made all data underlying the findings in their manuscript fully available?

Reviewer #1: Yes

Reviewer #2: Yes

5. Is the manuscript presented in an intelligible fashion and written in standard English?

Reviewer #1: Yes

Reviewer #2: Yes

6. Review Comments to the Author

Reviewer #1: In my opinion, the authors have taken into account all suggestions provided by the reviewers, including mine. They have also adequately explained when choosing not to.

I recommend this version por publication in the journal.

Reviewer #2: (No Response)

7. PLOS authors have the option to publish the peer review history of their article (what does this mean?). If published, this will include your full peer review and any attached files.

Reviewer #1: **Yes: **Vivienne C. Bachelet

Reviewer #2: No

---

## [Editor Report · Acceptance letter]

13 Oct 2021

PONE-D-21-18264R1 

Knowledge syntheses in medical education: Meta-research examining author gender, geographic location, and institutional affiliation 

Dear Dr. Maggio:

I'm pleased to inform you that your manuscript has been deemed suitable for publication in PLOS ONE. Congratulations! Your manuscript is now with our production department. 

Kind regards, 

on behalf of

Dr. Boris Bikbov 

Academic Editor

PLOS ONE